# Behavioral dissection of hunger states in *Drosophila*

Kristina J Weaver[1]*, Sonakshi Raju[2], Rachel A Rucker[3], Tuhin Chakraborty[1], Robert A Holt[2], Scott D Pletcher[1]*

[1]Department of Molecular and Integrative Physiology and Geriatrics Center, Biomedical Sciences and Research Building, University of Michigan, Ann Arbor, United States; [2]College of Literature, Science, and the Arts, Biomedical Sciences and Research Building, University of Michigan, Ann Arbor, United States; [3]Neuroscience Graduate Program, University of Michigan, University of Michigan, Ann Arbor, United States

**\*For correspondence:**
kjweaver@umich.edu (KJW);
spletch@umich.edu (SDP)

**Abstract** Hunger is a motivational drive that promotes feeding, and it can be generated by the physiological need to consume nutrients as well as the hedonic properties of food. Brain circuits and mechanisms that regulate feeding have been described, but which of these contribute to the generation of motive forces that drive feeding is unclear. Here, we describe our first efforts at behaviorally and neuronally distinguishing hedonic from homeostatic hunger states in *Drosophila melanogaster* and propose that this system can be used as a model to dissect the molecular mechanisms that underlie feeding motivation. We visually identify and quantify behaviors exhibited by hungry flies and find that increased feeding duration is a behavioral signature of hedonic feeding motivation. Using a genetically encoded marker of neuronal activity, we find that the mushroom body (MB) lobes are activated by hedonic food environments, and we use optogenetic inhibition to implicate a dopaminergic neuron cluster (protocerebral anterior medial [PAM]) to α'/β' MB circuit in hedonic feeding motivation. The identification of discrete hunger states in flies and the development of behavioral assays to measure them offers a framework to begin dissecting the molecular and circuit mechanisms that generate motivational states in the brain.

## eLife assessment

This **important** paper advances our ability to understand feeding behavior in fruit flies and begins to address the challenging question of motivation. With innovative methods based on the detailed monitoring of interactions between foods of different qualities at different hunger states, they present **compelling** evidence for non-homeostatic feeding not driven by metabolic need.

## Introduction

Hedonist philosophers believed that pleasure is the supreme goal of life and that behavior is determined by the pursuit of it (*Aristotle, 2004*). Hunger for tasty delicacies, for example, is a particularly potent motivator of behavior, as it can be promoted not only by a physiological need to consume nutrients, but also by the hedonic properties of food. Due to its dichotomous nature, defining behavioral and neural correlates of hunger is a challenge that has perplexed scientists for over a century. Early physiologists wondered how to explain hunger and where it might originate, and they eventually described hunger as a dull ache or gnawing sensation arising from the stomach or abdominal region (*Cannon and Washburn, 1993*). This historical description has aided the discovery of hormones like ghrelin, insulin, and others that communicate information from the periphery to the brain and have

roles in controlling feeding in response to energetic or nutritional needs, a state we now refer to as homeostatic hunger. In the decades since these discoveries, it has become clear that peripheral signals and others are integrated in the central nervous system to promote changes in behavior and reward signaling, which we interpret as indicators of hunger, but the nature of the motivational brain systems that drive these outputs remains less straightforward to identify and quantify.

Homeostatic feeding, or 'need-based' intake, is defined as food intake that is necessary to ensure caloric and nutritional needs are met, and it has been quantified by measuring bulk food intake. Such measures may be of total caloric intake or consumption of specific nutrients, but it is unclear whether these measures are representative of hunger that is driven by energetic need, nutrient deficiency, or some other motivation, which has led to variation in the way measures of homeostatic hunger are reported and interpreted (for review of mammalian brain circuits that modulate feeding, see *Alcantara et al., 2022*; *Andermann and Lowell, 2017*; *Saper et al., 2002*). However, the observation that most animals, including humans, regulate food intake around a protein target and the identification of discrete neuronal populations that regulate protein-specific feeding in model systems suggest that levels of homeostatic hunger may be primarily determined by the amount of protein an animal requires and thus, in recent years, protein intake has become a common method for measuring need-based hunger (*Gosby et al., 2011*; *Liu et al., 2017*; *Ro et al., 2016*; *Simpson et al., 2003*; *Weaver et al., 2023*).

Our understanding of hunger has continued to evolve, due in part to the observation that the pleasurable effects of food also stimulate feeding independently of physiological need, a state referred to as hedonic hunger (*Lowe and Butryn, 2007*). Hedonic, or 'need-free', intake is thought to be evoked by the incentive salience or tastiness of food, and thus, the sweetness of the food environment is predicted to play a larger role (*Lowe and Butryn, 2007*). Indeed, hedonic hunger is likely the predominant driver of feeding in modern human environments where food is readily available and energy deprivation is seldom experienced. Hedonic responses to various food or environmental manipulations have been described as either 'liking' responses (relating to the pleasure of a rewarding experience) or 'wanting' responses (relating to the motivation to seek a reward), which arise from different neuronal substrates (reviewed in *Berridge, 2009*). Accordingly, 'liking' is measured in rats by tracking facial responses in response to sweet tastants, while 'wanting' is measured by tracking latency to search for and consume a tasty food treat or by using the novelty suppressed feeding assay that attempts to measure an animal's ability to overcome an aversive environment to consume a food reward (*Arcego et al., 2020*; *Peciña et al., 2006*). While observations like these have greatly aided our ability to identify hedonic hunger states in mammals, much less is known about their neuronal and mechanistic origins relative to homeostatic hunger.

Although we now understand that hunger stratifies into two distinct states, one driven by need and the other by pleasure, we have a limited understanding of their behavioral and neuronal distinctions and connections. In mammals, brain circuits that modulate homeostatic or hedonic appetites have been identified. The well-described AGRP neurons in the arcuate nucleus of the hypothalamus, for example, are critical components of a circuit for homeostatic feeding (for review, see *Sternson and Eiselt, 2017*). They express receptors for peripherally released hormones that signal energetic state, are more active upon food deprivation, and transmit a negative valence signal that drives animals to find and consume food when the need arises (*Betley et al., 2015*; *Denis et al., 2015*; *Willesen et al., 1999*; *Zigman et al., 2006*). In addition to systems that seem to motivate homeostatic feeding, distinct circuits have also been associated with hedonic feeding including the monoaminergic populations that express serotonin and/or dopamine, and the mesolimbic dopamine pathway (*Ahn et al., 2022*; *Altherr et al., 2021*; *Denis et al., 2015*; *Ghiglieri et al., 1997*; *Rossi and Stuber, 2018*). While there is evidence to suggest that both branches of hunger are modulated by homeostatic state and hedonic feedback (e.g., *Denis et al., 2015*; *Krashes et al., 2011*; *Zigman et al., 2006*), the mechanistic origins that determine how various hunger drives work together and independently to generate feeding responses have yet to be fully described.

Simpler model systems, such as the fruit fly, *Drosophila melanogaster*, have proven valuable for insights into the neural circuitry that modulates feeding, but the nature of their hunger drives is unknown. Whether flies experience hedonic hunger, for example, is unclear, although it seems likely that they do; hungry flies learn to associate odors with a sucrose reward and when fed a high-sugar diet they overeat and become obese (*Huetteroth et al., 2015*; *May et al., 2019*). These behaviors

are all thought to occur through mechanisms involving dopaminergic neurons in the fly brain which, when artificially activated, promote foraging and food intake (*Tsao et al., 2018*). Honeybees have also exhibited signatures of food wanting, a component of hedonic hunger, which manifests through activation of dopaminergic populations in the brain that are reflective of reward expectation (*Huang et al., 2022*).

To better understand the different motive forces that drive feeding, it seems clear that there is a need to move beyond measures of food intake or foraging behavior, which have been useful for identifying external and internal factors that change the way an animal eats, in order to dissect when and to what degree animals may be experiencing different types of hunger. Here, we describe our first efforts to address this need by generating clear behavioral definitions of hunger states in *Drosophila* with the hope of providing an attractive system for defining the molecular features of their independent roles and the relationships between them. We characterize a repertoire of feeding micro-behaviors in the fly and use them to argue that *Drosophila* experience both homeostatic and hedonic hunger. We identify neural correlates of hedonic hunger in specific lobes of the *Drosophila* mushroom body (MB) and show that they are required for hedonic feeding. Finally, we develop a framework to examine the relationships between hedonic and homeostatic hunger states and present evidence for the existence of a system that acts as a potentiator of the relationship between them. Translating findings about the nature of feeding drives and the relationships among them could have far-reaching implications for obesity and aging research, and other behaviors influenced by motivational drives, including drug addiction and eating disorders.

## Results

We set out to complement and enrich traditional measures of feeding by identifying new behavioral metrics that might be useful for classifying different hunger states. Fly feeding behavior is typically described as changes in volumetric food intake, which is difficult to accurately quantify at the individual level due to the small amount of food ingested (*Deshpande et al., 2014*; *Shell et al., 2021*; *Shell et al., 2018*), or by using automated systems that detect physical interactions with food sources (*Itskov et al., 2014*; *Ro et al., 2014*). These automated devices have revealed rhythmic patterns of feeding behavior in flies that mimic feeding strategies in other animals, suggesting that the behaviors are highly organized and centrally regulated. They are, however, unable to distinguish different stereotypical actions that a freely moving fly might exhibit when interacting with a food source. Detailed behavioral characterizations of this sort have proven to be useful read-outs of neural states in other contexts and have the potential to inform our understanding of the motivational states that drive feeding (*Chen et al., 2002*; *Duistermars et al., 2018*; *Ning et al., 2022*). Much like human infants, for example, facial expressions and tongue movements in rats in response to appetitive tastants have been used to identify hedonic reactions and the brain systems that mediate them (*Berridge, 1996*).

To identify nuances in individual fly feeding, we developed video systems for manual and high-throughput annotation of what flies do when they eat. In all cases, we recorded freely moving flies placed in feeding chambers of the Fly Liquid-Food Interaction Counter (FLIC) system, which is designed to continuously record individual flies' interactions with food sources (*Ro et al., 2014*). We began by simply recording flies in the FLIC and manually curating each video in its entirety (*Figure 1A*). With the hope of observing the greatest variety of behaviors, we collected observations from flies ranging from 5 to 20 days old, from mated female and male sated and starved flies, from flies housed singly or in groups of up to three, and from flies interacting with 5% or 10% sucrose food. We observed 30 flies and nearly 300 'events', which were demarcated by intervals greater than 1 s during which the fly was not in physical contact with the food. Events were easy to identify, and different behavioral characteristics associated with events were easy to distinguish from each other (*Slooten, 1994*). Rare events (observed in fewer than 5% of flies) were not considered further, and events during which sight of the fly was out of focus or obscured (typically 1–2 events per fly) were ignored. Based on the remaining events, we created a library of user-defined behaviors that captured common, distinct, and definable interactions of flies with their food (*Table 1*).

We classified the behaviors we observed into five main categories: two non-feeding behaviors ('other' and 'interaction' events) and three feeding behaviors ('fast' feeding events, 'long' feeding events, and 'long/quick' feeding events) (*Table 1*). The first feeding event type comprised fast feeding events (termed 'F' or 'FI') during which we observed contact of the proboscis with the food for less

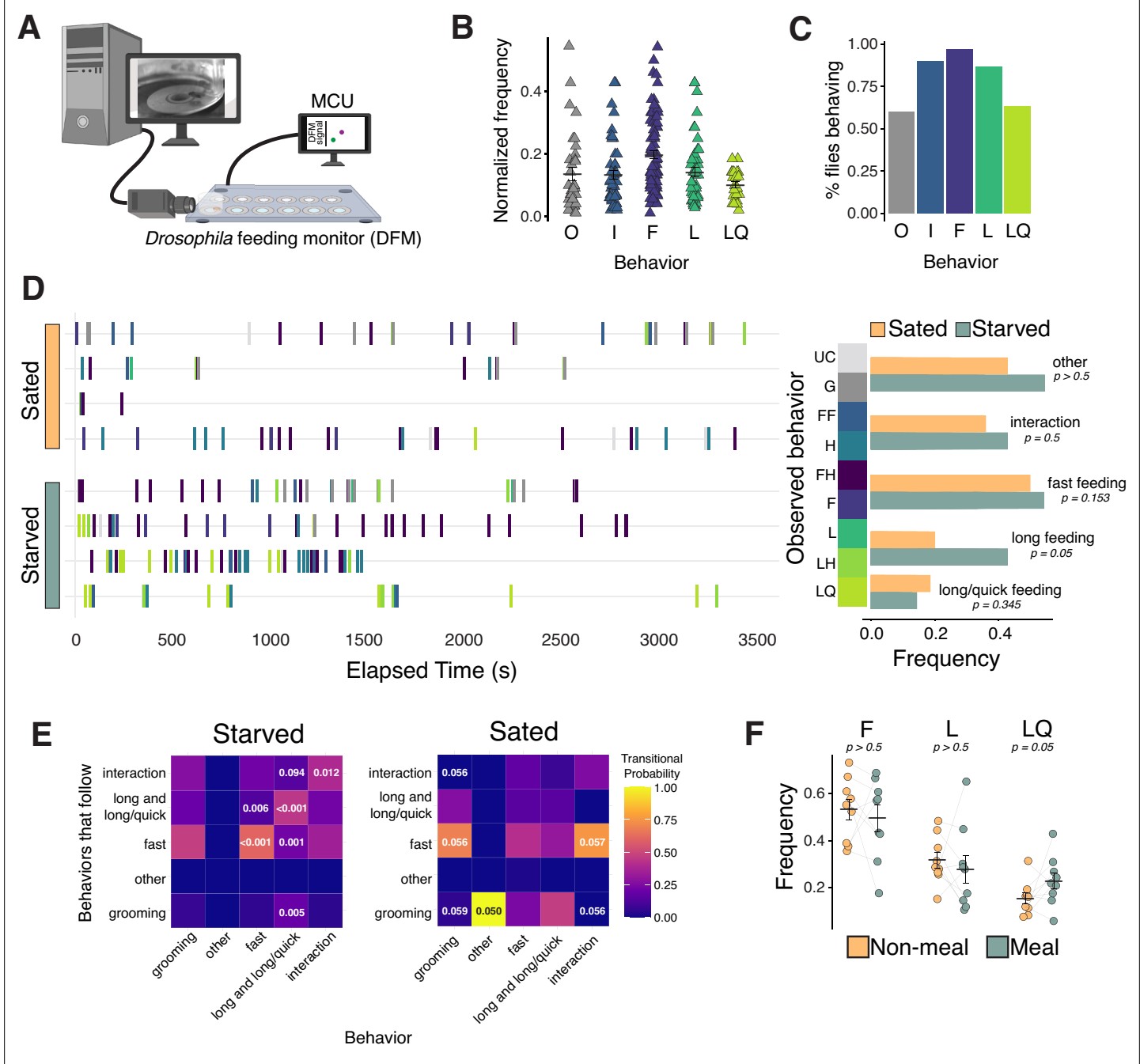

**Figure 1.** Characterization of feeding micro-behaviors in flies. (**A**) Cartoon schematic of setup for filming flies on the Fly Liquid-Food Interaction Counter (FLIC). Frame capture is triggered by interactions of flies with the food source. (**B**) Frequency of all five main categories of observed feeding micro-behaviors: other (**O**), interaction (**I**), fast (**F**), long (**L**), long/quick (**LQ**). Each data point represents the total number of times an event occurred normalized to all observed events for each fly. N=8 flies, 528 events. (**C**) Percentage of flies from (**B**) that engaged in each feeding micro-behavior. (**D**) Representative temporal plots of sated (N=4 flies, 171 events) and starved (N=4 flies, 357 events) feeding micro-behaviors (left panel) and frequency of each behavior (right panel, two-tailed t-tests). Each row in left panel represents one fly, and frequencies are relative to all behaviors (two-tailed t-tests). (**E**) Heat maps of starved (left) and sated (right) transitional probabilities of observed behaviors from (**D**) (transitional probabilities are generated by dividing the observed count for each event pair by the total occurrences of the given event. p-Values are determined based on Z-scores, as described in ***Blumstein and Daniel, 2007***). Long events contain both L and LQ events. (**F**) Frequency of each feeding micro-behavior during meal (7AM/PM-10AM/PM) vs. non-meal-times of day (one-tailed t-tests). N=9 flies, 693 events, frequency is relative to total interactions+feeding events. All error bars represent the mean +/- SEM.

The online version of this article includes the following source data and figure supplement(s) for figure 1:

*Figure 1 continued on next page*

*Figure 1 continued*

**Source data 1.** Excel spreadsheet containing source data used to generate Figures 1B-D.

**Figure supplement 1.** 24 hr FLIC recording demonstrating distinct morning and evening feeding peaks.

than 4 s. These types of events were common (occurring in ~95% of flies, *Figure 1B and C*) and were sometimes simultaneous with the interaction of a fly's legs with the food. Longer feeding events (termed 'L', 'LI', or 'LQ') were also identified, during which time we observed contact of the proboscis with the food for more than 4 s. Long events were less common than F-type events and could also include simultaneous interaction of legs with food. Long feeding events were either a continuous and sustained interaction of the proboscis with the food ('L' and 'LI', occurring in ~85% of flies, *Figure 1B and C*) or, less commonly, a continuous interaction during which the proboscis moved quickly in and out of the food ('LQ', occurring in ~60% of flies, *Figure 1B and C*). The first type of non-feeding behavior was termed an interaction event ('IF' or 'IH') during which the fly interacted with the food only using its legs; no proboscis contact was observed. These putative tasting interactions often occurred in a 'pitter-patter' motion, as if the fly were 'playing' with their food, and they occurred in greater than 90% of flies (*Figure 1B and C*). Lastly, we occasionally observed other non-feeding events ('G' or 'UC') that, nevertheless, often occurred just before or after feeding (occurring in ~60% of flies, *Figure 1B and C*). These tended to be various grooming behaviors, including using the forelegs to rub the head after interacting with food.

We next asked whether our behavioral classification could be used to distinguish hungry from sated flies. Recognizing a need to significantly increase the number of observed feeding events to provide sufficient statistical power to detect differences in behavioral tendencies, we designed a direct communication link between our custom DTrack video software and the FLIC system to trigger video capture only when there was physical contact between a fly and the food source, together with 5 s windows before and after the contact. This allowed us to create video compilations of feeding behaviors for fully fed or starved (24 hr) flies over several hours. Over 500 distinct events in these compilations were then manually scored using our classification of feeding behaviors, and comparisons were made between fully fed and starved flies for each behavioral category in *Table 1*. We observed that starved flies exhibited a greater frequency of feeding behaviors overall, as expected (sated event frequency = 0.606 events/min compared to starved event frequency = 0.788 events/min, p=0.061, data not shown), and that the relative frequency with which L-type feeding events occurred was increased (*Figure 1D*, quantified at right). The observation that starved flies exhibit more frequent

**Table 1.** Library of feeding micro-behaviors.

| Main event type | Code | Behavior description |
|---|---|---|
| Other (O) | G | Groom |
| | UC | Unknown contact |
| Interaction (I) | IF | Front legs touch food |
| | IH | Hind legs touch food |
| Fast feeding (F) | F | Single movement of proboscis into food to feed for 1–3 s |
| | FI | Meets requirements for F and I |
| Long feeding (L) | L | Proboscis continuously in food while feeding for >4 s |
| | LI | Meets requirements for L and I |
| Long/quick feeding (L) | LQ | Proboscis moves in and out quickly while feeding for >4 s |

long-duration events is consistent with recent reports that starved flies increase 'sips per burst' – a measure that is similar to our definition of L and LQ events (*Itskov et al., 2014*).

Qualitative aspects of the relationships between individual behaviors – that is, the architecture that results from sequences of behaviors – can reveal complexities that may also distinguish between different neural states (*Casarrubea et al., 2019*; *Casarrubea et al., 2015*; *Flavell et al., 2022*). Thus, we asked whether certain sequential patterns in fly feeding behavior appear repeatedly and whether they are affected by starvation. To do this, we used classic sequence analysis to map the probability that one event follows another in a sequence more often than what is expected to occur by chance alone – an approach that has been useful for identifying stereotyped patterns or structures of behaviors that tend to occur together (*Ning et al., 2022*; *Slooten, 1994*). We observed that hungry flies, whom we had starved for 24 hr, but not sated flies, followed F feeding events with additional F feeding events (transitional probability = 0.6 in starved flies with p-value <0.001, compared to a non-significant transitional probability = 0.55 in fully fed flies) and L-type feeding events with additional L-type feeding events (transitional probability = 0.439 in starved flies with p-value <0.001 compared to non-significant transitional probability = 0.1538 in fully fed flies) more often than would be expected by chance (*Figure 1E*). In contrast, fully fed flies were likely to engage in F-type events after grooming or interaction events and were more likely to groom following most other events, which may be reflective of a less goal-oriented behavioral state compared to that of starved flies (*Figure 1E*).

Having established that long-duration events occur more frequently and in a predictable sequence in starved flies, we sought to replicate this finding in flies during their normal circadian 'meal-time', when they are presumably physiologically hungry but not yet starved. When monitored over several days on the FLIC, flies exhibited circadian meal patterns that corresponded to the light cycle, with one peak in the morning when lights turn on and a second peak in the evening when lights turn off (e.g., *Figure 1—figure supplement 1*). We obtained 24 hr video compilations of nearly 700 feeding events in the FLIC and used the flies' endogenous feeding rhythm to designate 'meal' and 'non-meal' times of day. We manually scored the videos during each time for feeding events (fast, long, and long/quick events). We found that during meal-times flies exhibited a significantly greater frequency of LQ events, while the frequencies of F and L events were statistically indistinguishable between times of day (*Figure 1F*). Thus, both starved flies and flies during their circadian meal-time exhibit an increase in the total number of feeding events together with an increase in the relative frequency of long feeding events.

The FLIC generates several quantitative signal measures when flies interact with food, and we asked whether any of these numerical metrics might substitute for direct visual observation and distinguish among fast feeding events, long feeding events, and interactions. To determine how signal measures change in accordance with a fly's behavior, we scored 15 video compilations (resulting in 332 independent events) of flies interacting on the FLIC for F-type, L-type, or I-type events, and we matched each visually identified behavior to its corresponding FLIC signal data (e.g., *Figure 2A–C*). We evaluated differences in the characteristics of signal intensity for each event (i.e., its variance, minimum, maximum, average, and total) as well as the duration of the signal from each event. We found that only signal duration effectively distinguished long events from both fast events and interactions (*Figure 2D–I*). Thus, we established that longer signal durations from the FLIC are representative of L-type feeding events. We also noted that maximum signal intensity and the variance of the signal can distinguish interactions from both types of feeding events, which may be broadly useful to distinguish non-feeding interactions (i.e., tasting events) from feeding events (*Ro et al., 2014*).

## Development of an assay to identify homeostatic and hedonic feeding

Having established that hungry flies increase the duration of their feeding events as well as the total number of events (i.e., *Figure 3A*, top panel), we next asked if we might be able to identify distinct hunger drives in flies and, if so, whether our behavioral and quantitative metrics would distinguish between them. Recognizing that the original assays used for behavioral observation, in which flies were exposed to a single, homogenous food source, were likely insufficient to distinguish different types of feeding drives, we designed two types of food choice environments that we might expect would distinguish homeostatic from hedonic feeding. Homeostatic feeding drive in flies is thought to be measured by the extent to which flies choose to consume a yeast-containing food over one consisting of sucrose only, which we will refer to hereafter as a yeast preference (*Liu et al., 2017*; *Ro*

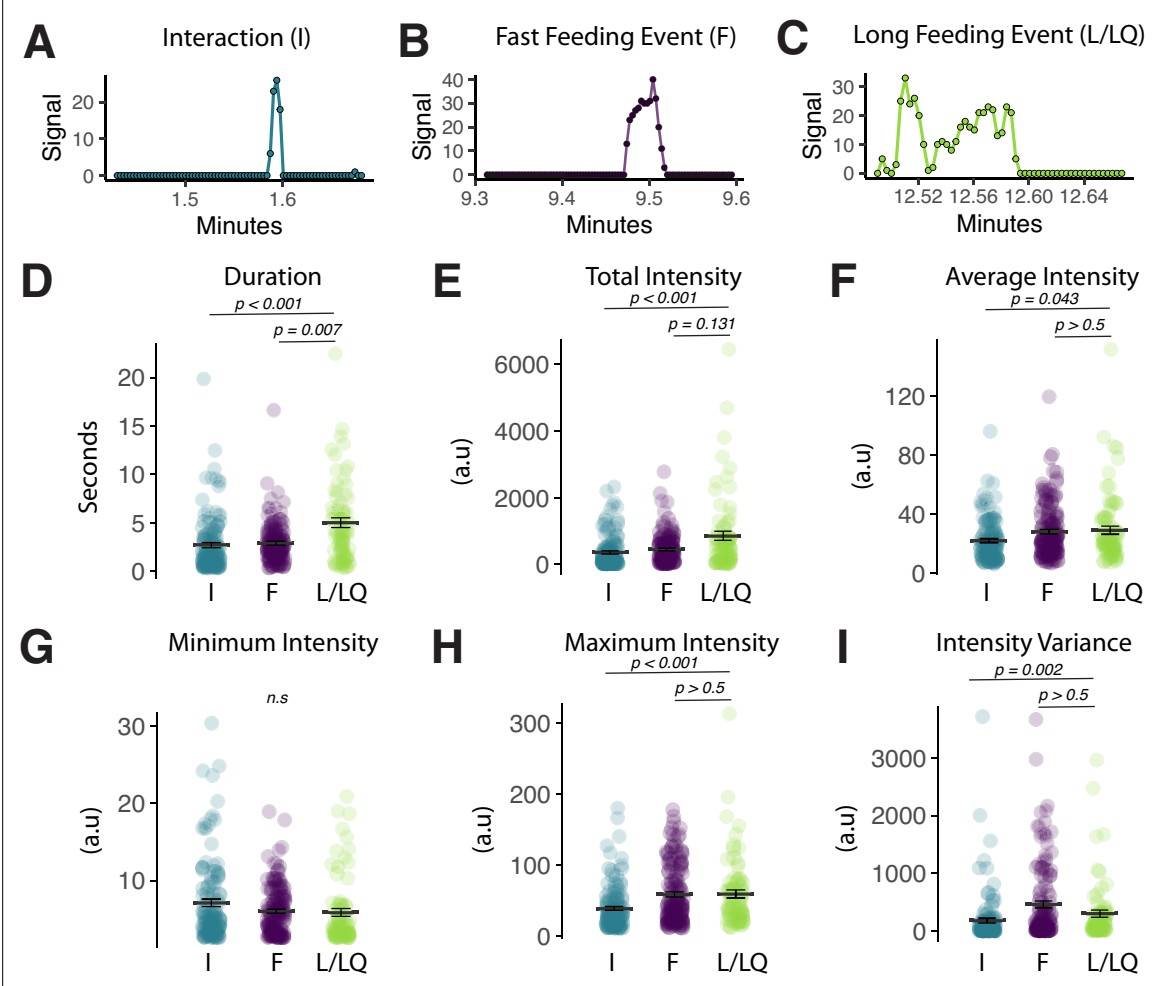

**Figure 2.** Fly Liquid-Food Interaction Counter (FLIC) signal characteristics distinguish between event types. (**A–C**) Representative FLIC signal generated during a visually identified interaction (**I**), fast feeding (**F**), or long feeding (**L/LQ**) event. (**D–I**) Signal characteristics generated by the FLIC during visually identified I, F, or L/LQ events. Each data point represents one event from one fly (one-way ANOVA with Tukey's post hoc). All error bars represent the mean +/- SEM.

The online version of this article includes the following source data for figure 2:

**Source data 1.** Excel spreadsheet containing source data used to generate Figures 2D-I.

*et al., 2016*). Therefore, for measures of homeostatic drive, we exposed flies to a choice of 2% sucrose vs. [2% yeast+1% sucrose], and we focused our analysis on the flies' relative interactions with the yeast containing food. To measure hedonic drive, on the other hand, we incorporated theorized requirements to experimentally measure hedonic hunger in mammals: animals must be energy replete and they must be tested on highly palatable food (*Lowe and Butryn, 2007*). We therefore exposed flies to a 'hedonic choice' environment which mimicked the 'homeostatic choice' environment except that the 2% sucrose food was replaced with sweeter 20% sucrose, and we focused our analysis on comparing interactions with the sucrose food between the food environments (e.g., *Figure 3A*, bottom panel). This design ensured flies remained sated by access to yeast while also allowing feeding on the highly palatable 20% sucrose solution.

We observed that the 2% sucrose environment elicited an increased yeast preference in starved compared to fully fed female flies (*Figure 3B*), reinforcing the notion that the total number of events with yeast vs. sucrose can be used as a relative measure of homeostatic hunger. We also observed an increase in yeast preference in fully fed female compared to male flies (*Figure 3C*), which may be reflective of a known difference in protein requirements, and therefore homeostatic hunger levels, between male and female flies. We next evaluated female and male fully fed flies in the 20% sucrose

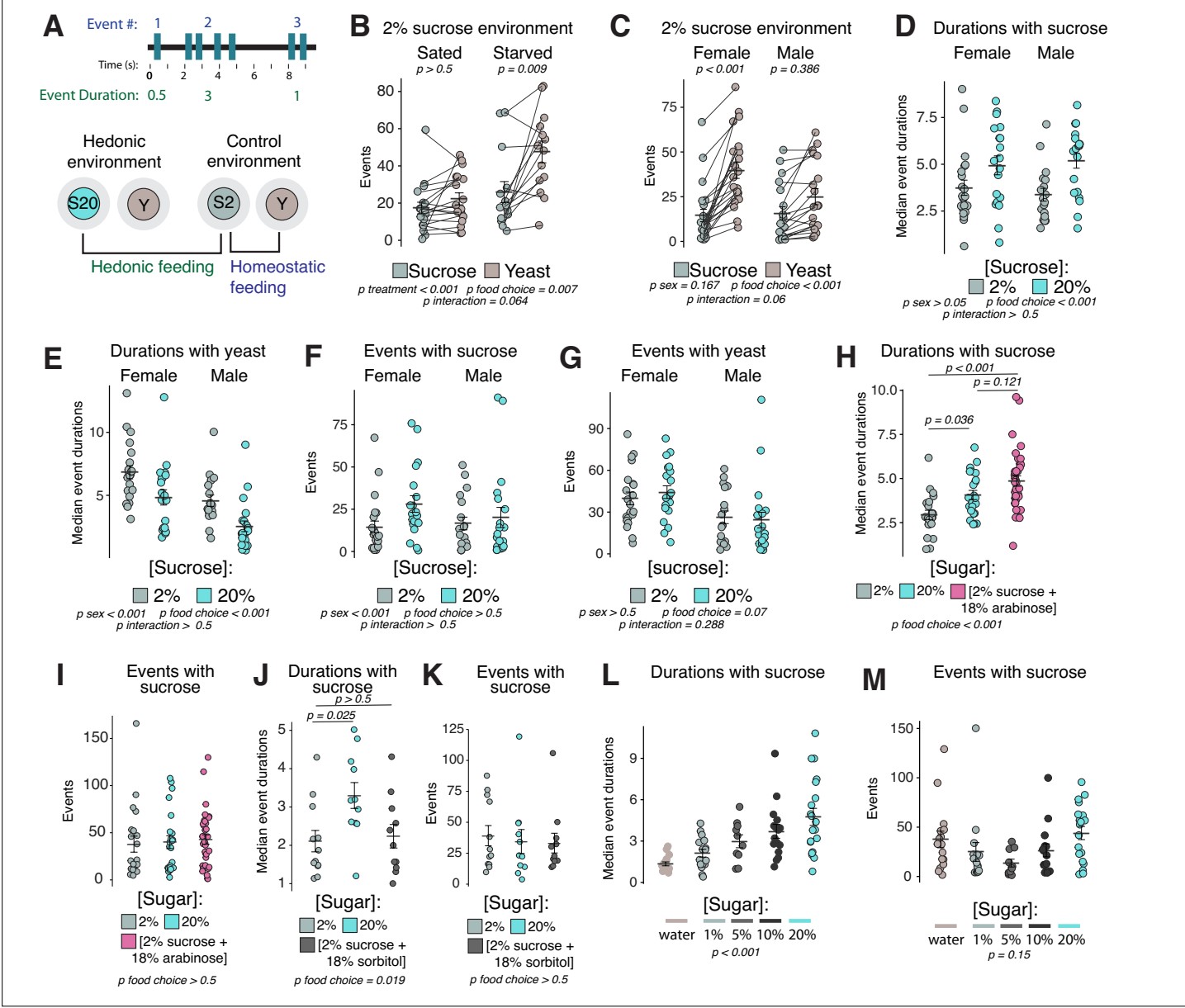

**Figure 3.** Feeding event durations are increased by hedonic food environments. (**A**) Cartoon schematic of event number and event duration (top). Cartoon schematic of hedonic and control food choice environments, and comparisons used to define metrics for homeostatic and hedonic feeding (bottom). (**B**) Total number of events with sucrose or yeast in the control food choice environment from sated or 24 hr starved female *Canton-S* flies (two-way ANOVA with Tukey's post hoc). (**C**) Total number of events with sucrose or yeast in the control food choice environment from sated female or male *Canton-S* flies (two-way ANOVA with Tukey's post hoc). (**D–E**) Median event durations with sucrose (**D**) or yeast (**E**) in the control vs. hedonic food choice environments from sated female and male *Canton-S* flies (two-way ANOVA). (**F–G**) Total number of events with sucrose (**F**) or yeast (**G**) in the control vs. hedonic food choice environments from sated female and male *Canton-S* flies (two-way ANOVA). (**H–K**) Median event durations (**H,J**) and total number of events (I,K) with sugar in the indicated food choice environments from sated female *Canton-S* flies (one-way ANOVA with Tukey's post hoc). (**J–K**) Median event durations (**J**) and total number of events (**K**) with sugar in the indicated food choice environment from sated female *Canton-S* flies (one-way ANOVA with Tukey's post hoc). (**L–M**) Median event durations (**L**) and total number of events (**M**) with sucrose in the indicated food choice environments (one-way ANOVA). All error bars represent the mean +/- SEM.

The online version of this article includes the following source data and figure supplement(s) for figure 3:

**Source data 1.** Excel spreadsheet containing source data used to generate Figures 3B-M and Figure 3 - Figure Supplement 1A-C.

**Figure supplement 1.** Hedonic food environments do not elicit increases in total number of events or event durations with yeast but do promote increased volumetric intake.

food environment compared to the 2% sucrose environment, which elicited increased event durations with 20% sucrose compared to the 2% sucrose food (*Figure 3D*) and also elicited decreased event durations with yeast food (*Figure 3E*). The total number of events on sucrose or yeast wells was statistically indistinguishable between food environments (*Figure 3F and G*), suggesting that modulation of event duration, but not total number of events, may be a feature of hedonic food environments. We also measured sucrose intake in the control vs. hedonic environments using a method that determines how much volume a group of flies consume in 24 hr (termed 'Con-Ex' for consumption-excretion) (*Shell et al., 2021*; *Shell et al., 2018*), and found that flies consume twice as much sucrose volume in the hedonic environment (*Figure 3—figure supplement 1A*). We concluded that the measurable increase in sucrose event durations in the hedonic food environment may reflect increases in a hedonic hunger drive while total number of events on yeast vs. sucrose food may reflect homeostatic state.

To challenge our measure of hedonic feeding, we reasoned that if increased event durations were indeed a result of hedonic hunger, then sweetness alone should be sufficient to increase event durations without modulating measures of homeostatic feeding. We tested two additional food choice environments where we altered either the sweetness or caloric content of the sugar choice. Sweetness was manipulated by using the non-metabolizable sweetener arabinose ([2% sucrose+18% arabinose]), which mimics the caloric and nutritive content of the 2% sucrose but contains the sweetness of the 20% sucrose choice. Caloric content was manipulated using the nutritive, but not sweet, sugar alcohol, sorbitol ([2% sucrose+18% sorbitol]), which mimics the caloric content of the 20% sucrose but contains the sweetness of the 2% sucrose (*Burke and Waddell, 2011*; *Hassett, 1948*). We observed that the sweet arabinose food environment elicited increased event durations that were comparable to flies in the 20% sucrose environment, with no statistical difference in total event number (*Figure 3H and I*, *Figure 3—figure supplement 1B*) and that the caloric but not sweet sorbitol food environment was not sufficient to increase event durations or total event number (*Figure 3J and K*). Finally, we tested several additional food choice environments containing a range of sucrose concentrations and observed that event durations were shortest when the sucrose choice contained only water or 1% sucrose and increased linearly as the sucrose concentration increased, while event durations with yeast food and total number of sucrose events were not statistically different (*Figure 3L and M*, *Figure 3—figure supplement 1C*). Thus, sweet taste was sufficient to evoke longer feeding event

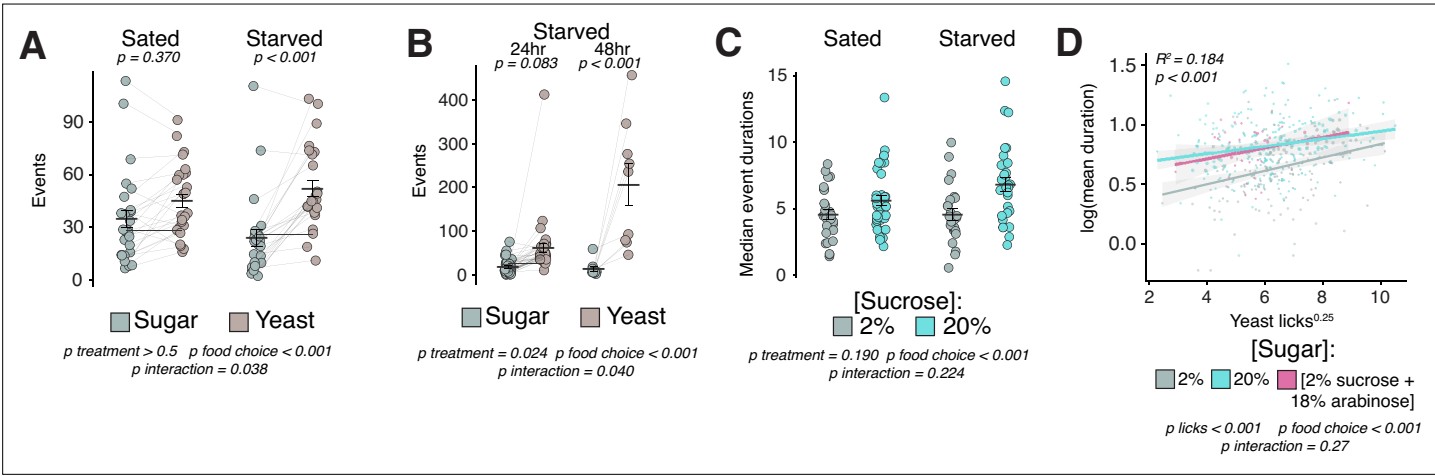

**Figure 4.** Hedonic feeding is modulated by homeostatic hunger. (**A**) Total number of events with sucrose or yeast in the control food choice environment from sated or 24 hr starved female *Canton-S* flies (two-way ANOVA with Tukey's post hoc). (**B**) Total number of events with sucrose or yeast in the control food choice environment from 24 hr vs. 48 hr starved female *Canton-S* flies (two-way ANOVA with Tukey's post hoc). (**C**) Median event durations with sucrose in the control vs. hedonic food environments from sated vs. 24 hr starved female *Canton-S* flies (two-way ANOVA). (**D**) Linear regression of interactions with yeast (licks) versus mean event durations with indicated sugar. Each data point represents one fly (yeast licks are expressed as a transformation to the 0.25 power and event durations are expressed as a log transformation to achieve normality, call = lm(SucroseDurations ~YeastLicks*FoodEnvironment), two-way ANOVA, adjusted $R^2$=0.1835). All error bars represent the mean +/- SEM.

The online version of this article includes the following source data for figure 4:

**Source data 1.** Excel spreadsheet containing source data used to generate Figures 4A-D.

durations, while calories alone were not, indicating that this metric obtained from our hedonic food choice paradigm may reliably quantify hedonic feeding.

Whether and how homeostatic and hedonic feeding mechanisms interact or overlap is controversial, and thus, we asked whether our metrics of homeostatic and hedonic hunger are correlated among individual flies. To obtain flies with a range of homeostatic needs, we used fully fed and starved flies to obtain feeding measures in our control or hedonic food choice environments. Starvation elicited an increase in yeast interactions compared to 2% sucrose (*Figure 4A*) that was more pronounced in flies starved for 48 vs 24 hr (*Figure 4B*), which is consistent with our expectation for increased homeostatic drive. We found that starvation seemed to marginally increase differences in sucrose event durations in hedonic vs. control environments, indicating that homeostatic hunger may enhance hedonic feeding (*Figure 4C*). For each fly, we next calculated metrics for homeostatic drive (# interactions with yeast food) as well as hedonic drive (sucrose event durations) and plotted their relationship along the x- and y-axes, respectively. We reasoned that if the circuits or programs that drive hedonic and homeostatic feeding were completely independent, then we should observe a slope close to zero. A significantly positive or negative slope, however, would suggest that increasing physiological need might also modulate hedonic hunger, or vice versa. We observed a significantly positive correlation between our measures of homeostatic and hedonic feeding (*Figure 4D*), suggesting that increasing physiological need might also potentiate hedonic hunger, or vice versa, which is consistent with the observation in mammals that hunger increases the reward value of food (*Fulton, 2010*). Interestingly, we also noted that separating regression lines from flies in the control vs. hedonic food environments revealed parallel lines whose slopes were not significantly different but were separated by a shift along the y-axis, indicating again that a sweeter food environment promotes longer event durations but may not modulate the relationship between hunger drives. These findings indicate that homeostatic state is a potentiator of hedonic feeding and establish that inferences of hunger states and measures of feeding should include considerations of each component independently as well as their interaction.

## Identification of neural correlates of hedonic hunger

A good deal is known about neural circuits associated with homeostatic hunger in *Drosophila*, and we wondered whether our behavioral and quantitative classification metrics could aid in identification of representative neural correlates of a hedonic hunger state. We began by conducting an unbiased screen for neurons that respond to hedonic food environments using the genetically encoded calcium indicator CaMPARI driven by a pan-neuronal GAL4 driver (Nsyb-GAL4). CaMPARI is a green fluorescent protein that irreversibly photoconverts to red when flies are simultaneously exposed to increases in intracellular calcium and UV light provided by the experimenter (*Edwards et al., 2020*; *Fosque et al., 2015*). We placed groups of three naïve, fully fed flies in the hedonic food choice environment or the control food environment and allowed them to habituate for 1 hr. Flies in each chamber were then exposed to cycles of UV light (10 s on/10 s off) for a total of 1 hr to label neuronal populations that were differentially active when flies are in the hedonic vs. the control food environment. Brains were dissected and imaged immediately, and the ratio of red:green fluorescent protein was analyzed, with a higher ratio indicating increased neuronal activity. We observed that the MB had a higher ratio of red:green fluorescence when flies were in the hedonic food environment, indicating that this population was more active (*Figure 5A*).

To validate the findings of our unbiased screen and as a direct test of MB activity in hedonic food environments, we restricted the expression of CaMPARI to the MB using the *238Y* enhancer-trap line (*Yang et al., 1995*). We observed that the hedonic food environment increased red:green fluorescence in the horizontal MB lobes compared to the control food environment (*Figure 5B*). The *Drosophila* MB is a well-described structure known to be involved in reinforcement and motivational control and is innervated by dopaminergic populations that are thought to encode reinforcing properties of food, including relaying information about sweet taste (*Aso et al., 2009*; *Aso et al., 2014*; *Das et al., 2016*). Thus, our observation that MB neurons are activated in hedonic food environments is consistent with prior knowledge of their roles in motivation and reward (*Landayan et al., 2018*).

To determine whether MB neurons are involved in hedonic hunger beyond correlative increases in activity, we tested for their requirement in hedonic and homeostatic feeding. We used the genetically encoded optogenetic tool GtACR to selectively inhibit MB neurons in our food choice assay. GtACR is

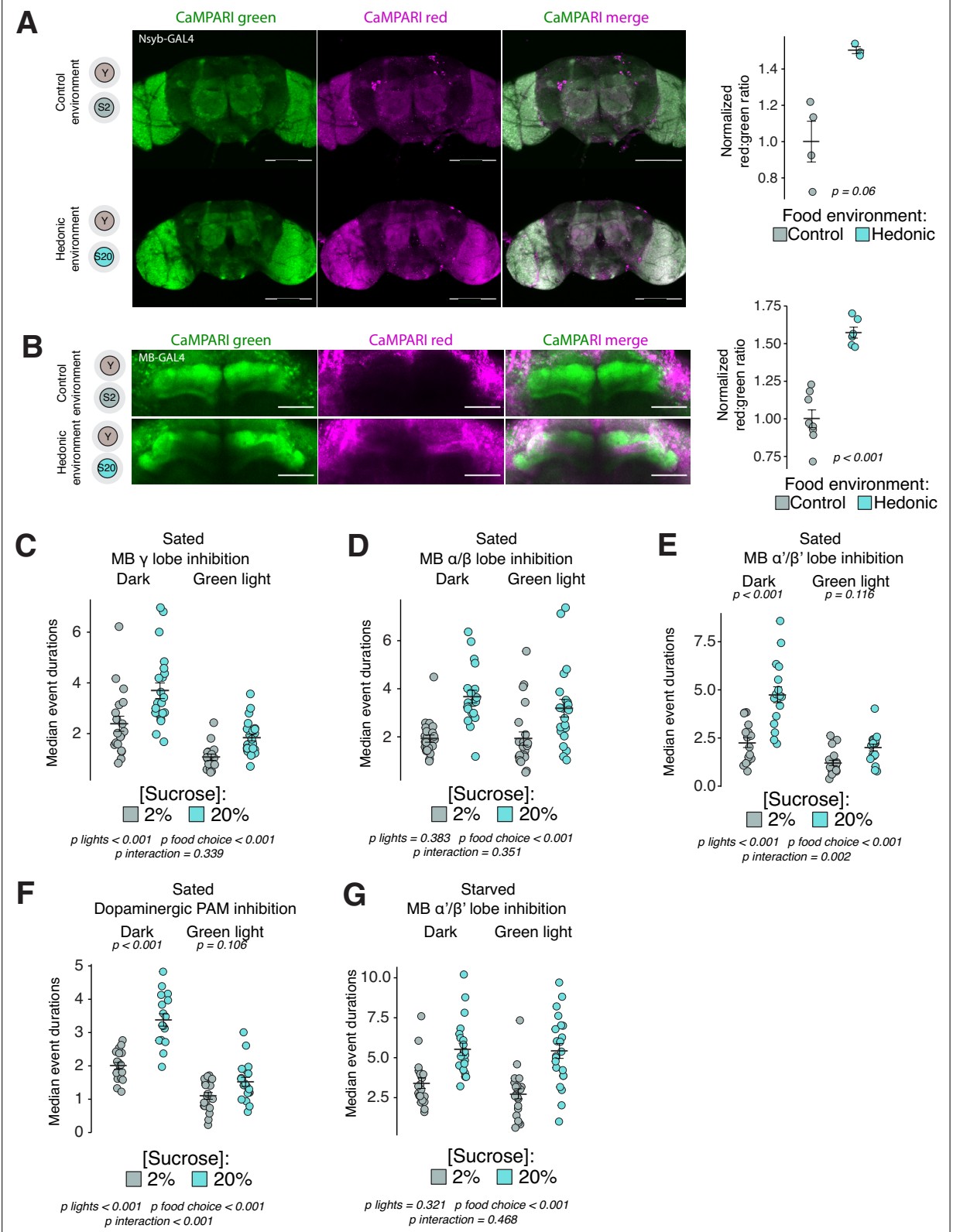

**Figure 5.** Hedonic hunger requires distinct mushroom body lobes. (**A**) Representative maximum intensity projections from brains of flies expressing *Nsyb-GAL4>UAS-CaMPARI* exposed to control or hedonic food environments, quantified at right (scale bar = 100 μM, two-tailed t-test). (**B**) Representative maximum intensity projections from brains of flies expressing *MB238Y-GAL4>UAS-CaMPARI*, quantified at right (scale bar = 10 μM, one-tailed t-test). For both (**A**) and (**B**), green represents unconverted CaMPARI and magenta represents converted CaMPARI. (**C–E**) Median event durations

*Figure 5 continued on next page*

*Figure 5 continued*

with sucrose in control vs. hedonic food environments from sated female flies expressing *UAS-GtACR* driven by either (**C**) *MB131B-GAL4* (γ(d);γ(m)), (**D**) *MB008B-GAL4* (α/β(c);α/β(p);α/β(s)), or (**E**) *MB461B-GAL4* (α′/β′(ap);α′/β′(m)) (two-way ANOVAs with Tukey's post hoc). (**F**) Median event durations with sucrose in control vs. hedonic food environments from sated female flies expressing *PAM-GAL4>UAS-GtACR* (two-way ANOVA with Tukey's post hoc). (**G**) Median event durations with sucrose in control vs. hedonic food environments from 24 hr starved female flies expressing *MB461B-GAL4>UAS-GtACR* (two-way ANOVA). All error bars represent the mean +/- SEM.

The online version of this article includes the following video, source data, and figure supplement(s) for figure 5:

**Source data 1.** Excel spreadsheet containing source data used to generate Figures 5A-G and Figure 5 - Figure Supplements 1A-C.

**Figure supplement 1.** Mushroom body lobes are not required for homeostatic hunger.

**Figure 5—video 1.** NsybCaMPARI stacks control environment green, related to *Figure 5A–B*.
https://elifesciences.org/articles/84537/figures#fig5video1

**Figure 5—video 2.** NsybCaMPARI stacks control environment red.
https://elifesciences.org/articles/84537/figures#fig5video2

**Figure 5—video 3.** NsybCaMPARI stacks control environment merge.
https://elifesciences.org/articles/84537/figures#fig5video3

**Figure 5—video 4.** NsybCaMPARI stacks hedonic environment green.
https://elifesciences.org/articles/84537/figures#fig5video4

**Figure 5—video 5.** NsybCaMPARI stacks hedonic environment red.
https://elifesciences.org/articles/84537/figures#fig5video5

**Figure 5—video 6.** NsybCaMPARI stacks hedonic environment merge.
https://elifesciences.org/articles/84537/figures#fig5video6

**Figure 5—video 7.** MBCaMPARI stacks control environment green.
https://elifesciences.org/articles/84537/figures#fig5video7

**Figure 5—video 8.** MBCaMPARI stacks control environment red.
https://elifesciences.org/articles/84537/figures#fig5video8

**Figure 5—video 9.** MBCaMPARI stacks control environment merge.
https://elifesciences.org/articles/84537/figures#fig5video9

**Figure 5—video 10.** MBCaMPARI stacks hedonic environment green.
https://elifesciences.org/articles/84537/figures#fig5video10

**Figure 5—video 11.** MBCaMPARI stacks hedonic environment red.
https://elifesciences.org/articles/84537/figures#fig5video11

**Figure 5—video 12.** MBCaMPARI stacks hedonic environment merge.
https://elifesciences.org/articles/84537/figures#fig5video12

a channelrhodopsin that is activated upon green light exposure and increases chloride conductance, resulting in hyperpolarization and silencing of neuronal populations (*Mohammad et al., 2017*). We expressed GtACR in either the γ, α/β, or α′/β′ lobes of the MB (*Aso et al., 2014*) and found that communication from the γ and α/β lobes is dispensable for both hedonic and homeostatic feeding (*Figure 5C and D*, *Figure 5—figure supplement 1A and B*). In contrast, signaling specifically from the α′/β′ lobes was required for hedonic, but not homeostatic, feeding (*Figure 5E*, *Figure 5—figure supplement 1C*). The MB horizontal lobes are innervated by dopaminergic neurons from the protocerebral anterior medial (PAM) cluster, which were also required for hedonic feeding (*Figure 5F*; *Mao and Davis, 2009*). These results indicate that a distinct PAM to α′/β′ MB circuit regulates hedonic feeding, which is consistent with known descriptions of anatomical and functional segregation of MB lobes (*Aso et al., 2014*; *Landayan et al., 2018*).

We next sought to determine whether communication from the α′/β′ lobe was required for the potentiating effects of homeostatic hunger on hedonic feeding. We reasoned that if the α′/β′ lobe also functions as a potentiator between the two hunger states, then inhibition of these neurons would be expected to prevent increases in hedonic feeding when flies are starved. Instead, we observed that starvation restored hedonic feeding in flies whose α′/β′ lobes were inhibited (*Figure 5G*), suggesting that potentiation of hedonic feeding by homeostatic state involves the recruitment of additional and independent circuits for hedonic feeding.

## Discussion

Animals exquisitely regulate feeding according to internal needs, but they are also subject to their desires. Thus, food intake only sometimes occurs in response to homeostatic hunger and can also occur in response to emotional stress, social environment, illness, and pursuit of reward (*Siemian et al., 2021*; *Xu et al., 2019*). Because animals eat for many reasons other than hunger, food intake alone may be a poor indicator of hunger states and may obscure investigations into the motivational forces that drive behavior. Here, we observed the feeding micro-behaviors of flies and developed a behavioral classification system to assist in the identification of feeding drives. We established that flies exhibit behavioral signatures of two hunger states, homeostatic and hedonic, and we used them to identify neurocircuits that are required for hedonic, but not homeostatic, feeding. We also showed that homeostatic and hedonic feeding are both independent and overlapping depending on environmental context, and we developed a simple interaction framework that researchers can use to investigate these relationships in detail. The development of this paradigm presents entry points into the discovery of new homeostatic and hedonic hunger circuits in flies, better classification of feeding circuits which have already been described, and dissection of mechanistic relationships among them.

The *Drosophila* MB supports a wide variety of behaviors and functions, in part due to its complexity and heterogeneity in anatomical structure (*Aso et al., 2009*). The lobes of the MB are formed by the interconnected axons of ~2000 Kenyon cells and 34 MB output neurons, and they receive modulatory input from ~20 dopaminergic cell types (*Aso et al., 2014*). The MB lobes have been anatomically and functionally divided into three classes: γ, α/β, and α′/β′, which have been further divided into functionally distinct compartments (*Aso et al., 2014*; *Crittenden et al., 1998*; *Tanaka et al., 2008*). Our data suggest that compartments of the α′/β′ MB lobes, but not the γ or α/β lobes, are required for hedonic feeding. Although our experiments isolated a discrete location within the functionally heterogenous MB, the α′/β′ lobe still contains more than 350 Kenyon cells that are divided into middle (m) and anterior-posterior (ap) compartments (*Aso et al., 2014*). Future investigations that utilize existing and emerging intersectional technologies will be useful for discerning the specific cells within the α′/β′ MB lobes that promote hedonic feeding, as well as their modulatory inputs and outputs, and for evaluation of the cell-intrinsic mechanisms that determine how hedonic hunger is encoded.

Connections from dopaminergic populations in the PAM cluster with MB-intrinsic Kenyon cells have been implicated in appetitive learning, signaling reward in response to sugar, motivation to overcome environmental stress, and promoting wakefulness (*Burke et al., 2012*; *Haynes et al., 2015*; *Hermanns et al., 2022*; *Liu et al., 2012*). PAM projections to Kenyon cell axons in the β′ compartment of the MB respond to sugar by increasing calcium transients, which has led to the idea that these synapses transmit information about sweet taste (*Liu et al., 2012*; *May et al., 2020*). Interestingly, these responses are increased twofold by starvation, suggesting that they scale in response to environmental context or degree of need (*Liu et al., 2012*), and artificial activation of PAM neurons also increases foraging behavior (*Tsao et al., 2018*). Our finding that the PAM>α′/β′ circuit is required for promoting hedonic hunger builds upon a rich body of literature, and together these observations offer a model that may unite the many described functions of this node. One possibility is that this circuit acts as a permissive or motive force that gates or fuels downstream mechanisms involved in behavioral responses to environmental stimuli. Indeed, learning, reward, and wakefulness are all processes that might benefit from or require a force that sustains them. This view offers an attractive entry into understanding the seat of motivation in the fly brain.

Decades of research on the neural control of feeding have resulted in the identification of mammalian neurocircuits involved in the control of feeding, and technological advances have allowed researchers to manipulate them with an increasing degree of specificity. We are, however, approaching the limits of what we can learn from manipulations and technologies of these sorts and thus, complimenting them with discoveries from model systems in which more specific manipulations can be tested may accelerate our ability to understand how the brain generates motivated states. The hedonic circuits we have identified here in flies, for example, can be immediately investigated at multiple levels to assess the neuronal and molecular mechanisms that determine their role in feeding, including spatio-temporal manipulation of cell-signaling pathways, electrophysiological recordings in live animals in response to various food environments or energetic deficits, and optogenetic manipulation paired with close behavioral monitoring. These types of experiments would fulfill several main goals in the study of the neural control of appetite regulation (*Ahn et al., 2022*) and, despite differences in the

precise identity of feeding neurocircuits between flies and mammals, discoveries about foundational principles and conserved signaling pathways may inform and direct mammalian studies.

# Materials and methods

**Key resources table**

| Reagent type (species) or resource | Designation | Source or reference | Identifiers | Additional information |
|---|---|---|---|---|
| Genetic reagent (*Drosophila melanogaster*) | *Canton-S* | Bloomington *Drosophila* Stock Center | | |
| Genetic reagent (*Drosophila melanogaster*) | *UAS-CaMPARI* | Bloomington *Drosophila* Stock Center | BDSC #68762 RRID:BDSC_68762 | |
| Genetic reagent (*Drosophila melanogaster*) | *MB238Y-GAL4* | Bloomington *Drosophila* Stock Center | BDSC #81009 RRID:BDSC_81009 | |
| Genetic reagent (*Drosophila melanogaster*) | *MB131B-GAL4* | Bloomington *Drosophila* Stock Center | BDSC #68265 RRID:BDSC_68265 | |
| Genetic reagent (*Drosophila melanogaster*) | *MB008B-GAL4* | Bloomington *Drosophila* Stock Center | BDSC #68291 RRID:BDSC_68291 | |
| Genetic reagent (*Drosophila melanogaster*) | *MB461B-GAL4* | Bloomington *Drosophila* Stock Center | BDSC #68327 RRID:BDSC_68327 | |
| Genetic reagent (*Drosophila melanogaster*) | *GMR58E02-GAL4* | Bloomington *Drosophila* Stock Center | BDSC #41347 RRID:BDSC_41347 | |
| Genetic reagent (*Drosophila melanogaster*) | *LexAop2-CsChrimson;UAS-CaMPARI2;GMR57C10-GAL4* | Bloomington *Drosophila* Stock Center | BDSC #81089 RRID:BDSC_81089 | |
| Genetic reagent (*Drosophila melanogaster*) | *UAS-GtACR* | Other | | M. Dus (University of Michigan) |
| Software, algorithm | RStudio | RStudio | RRID:SCR_000432 | |
| Software, algorithm | Fiji | ImageJ | RRID:SCR_002285 | |
| Software, algorithm | FLIC analysis R code | Flidea | RRID:SCR_018386 | |
| Software, algorithm | DTrack | Other | | S. Pletcher (University of Michigan) |
| Other | FLIC *Drosophila Feeding Monitors* | Sable Systems | Model DFMV3 | https://www.sablesys.com/products/classic-line/flic_drosophila_behavior_system/ |
| Other | FLIR grasshopper camera | FLIR | GS3-U3-28S5C-C | #88-052 - GS3-U3-28S5C-C 2/3" FLIR Grasshopper3 High Performance USB 3.0 Color Camera |
| Other | Fujinon varifocal lens | Fujinon | MFR #DV3.4x3.8SA-1 | Fujinon 3 MP varifocal lens (3.8–13 mm, 3.4× zoom) |

## Fly stocks and husbandry

Fly stocks were maintained on a standard cornmeal-based larval growth medium and in a controlled environment (21°C, 60% humidity) with a 12:12 hr light:dark cycle. We controlled the developmental larval density by manually aliquoting 32 µL of collected eggs into individual bottles containing 25–50 mL of food at 25°C. For FLIC experiments, following eclosion mixed sex flies were kept on cornmeal medium at 18°C until they were 7–14 days old. One day prior to the experiments, flies were sorted by sex and transferred onto either SY10 medium (10% [w/v] sucrose and 10% [w/v] yeast) or wet starved (1 Kimwipe with 2 mL of $H_2O$) at 25°C. Experiments that used flies starved for 48 hr were sorted to starvation vials and kept at 25°C 2 days prior to the experiments. We used mated

female and male flies that were between 7 and 14 days old for all video characterization experiments and mated female flies for all subsequent experiments, except where noted. The following stocks were used for experiments: *Canton-S, UAS-CaMPARI* (BDSC #68762), *MB238Y-GAL4* (BDSC #81009), *MB131B-GAL4* (BDSC #68265), *MB008B-GAL4* (BDSC #68291), *MB461B-GAL4* (BDSC #68327), *GMR58E02-GAL4* (BDSC #41347), and *LexAop2-CsChrimson;UAS-CaMPARI2;GMR57C10-GAL4* (BDSC #81089) were obtained from the Bloomington *Drosophila* Stock Center. *UAS-GtACR* was provided by M. Dus (University of Michigan, MI, USA).

## Video recording and scoring

The video recording setup for initial behavioral characterizations (*Figure 1B and C*, *Table 1*) consisted of the FLIC, a custom single-well enclosure, and a light to illuminate the recording area. A FLIR Grasshopper camera (GS3-U3-28S5C-C) fitted with a Fujinon varifocal lens (MFR #DV3.4x3.8SA-1) was positioned against the lid to record the FLIC well and the surrounding enclosed area. Video recording and FLIC data collection were started simultaneously, and video length was dependent on the maximum number of frames that could be stored. This setup recorded continuous video of the fly and saved this footage unaltered. For experiments in *Figure 1D–F*, we developed a direct communication link between our custom DTrack video software and the FLIC system to trigger video capture only when there was physical contact between a fly and the food source, together with 5 s windows before and after the contact (see *Figure 1A*).

Videos of sated or 24 hr wet-starved male and female Canton-S flies were recorded. A single fly was aspirated into the FLIC, which contained 5% or 10% sucrose food solution in 4 mg/L MgCl$_2$. Videos were recorded at various times of day between 10AM and 7PM and had an average length of about 51 min, except for videos collected for circadian analysis, which were recorded over a 24 hr period. 'Meal-times' were defined as 7AM-10AM and 7PM-10PM. All videos were recorded on the lab bench. Manual, frame-by-frame annotation was used to classify user-defined behaviors. Experimenters were blinded to the treatment for video scoring. Data was analyzed using the open-source JWatcher software and R studio.

## FLIC assays

Flies were tested on the FLIC system as previously described (*Ro et al., 2014*) (Sable Systems International). FLIC *Drosophila* Feeding Monitors (DFMs, Sable Systems International, model DFMV3) were used in the two-choice configuration and each chamber was loaded with food solutions containing a sugar choice (1–20% sucrose, [18% arabinose+2% sucrose], or [18% sorbitol+2% sucrose]) and a yeast choice ([2% bacto yeast extract+1% sucrose]) in 4 mg/L MgCl$_2$. Flies were briefly anesthetized on ice and aspirated into the DFM chambers, ensuring that each DFM received flies from all treatments. We began recording immediately after loading flies (generally, loading all DFMs requires <10 min) and measured FLIC interactions for 6 hr during the 'non-meal' time of day. Each FLIC experiment contains pooled data from two independent replicates. FLIC data were analyzed using custom R code, which is available at https://github.com/PletcherLab/FLIC_R_Code, (*Weaver, 2023* copy archived at swh:1:rev:9713cfeb88c26e7aa4c0d87b26fcf3361500c670). Default thresholds were used for analysis except for the following: minimum feeding threshold = 10, tasting threshold = (0,10). Animals that did not participate (i.e., returned zero values), whose DFM returned an unstable baseline signal, or who produced extreme outliers (criteria for outliers were predetermined as exceeding twice the mean of the population) were excluded from analysis.

## Con-Ex feeding assays

Con-Ex experiments were carried out as previously described (*Shell et al., 2021*; *Shell et al., 2018*). Experimental female flies were sorted to appropriate control (choice between SY10 food and 2% sucrose) or hedonic (choice between SY10 and 20% sucrose) food environments (10 flies per vial) when they were 4 days old and maintained on this food for 3 weeks. Food was changed every 2–3 days. After the dietary pretreatment period, blue test food was prepared in removable choice caps by adding 1% (w/v) FD&C Blue No. 1 to only the side of the cap containing the sucrose food choice. Flies were moved to fresh vials with the removable blue-food caps on the top of the vials and were allowed to feed and excrete for 24 hr. Caps and flies were removed after the 24 hr test period and flies were counted. Excreted dye was collected by vortexing each vial with 3 mL of water. Concentration

of the dye was determined by absorbance at 630 nm and compared to a standard curve of known concentrations.

## Optogenetic FLIC assays

Flies expressing UAS-GtACR in desired neuronal populations were reared as described above. When 7–14 days old, flies were sorted to cornmeal medium containing 200 μm all-*trans*-retinal (from a stock solution of 100 mM ATR in 100% ethanol) and kept in the dark for 1–4 days at 18°C. Flies were shifted to 25°C for 24 hr before the experiment and flipped to either SY10 media containing 800 μm all-*trans*-retinal or starvation vials. Custom FLIC lids were fitted with 530 nm LEDs (Luxeon). Custom hardware and firmware were designed to allow the experimenter to control LED intensity and a range of light stimulus parameters. We used a stimulus frequency of 40 Hz and a pulse width of 800 ms for the entire duration of the 6 hr FLIC experiment.

## CaMPARI

CaMPARI photoconversion was carried out according to previously established protocols with some modifications (*Edwards et al., 2020*). A 405 nM LED (Thorlabs M405L3) was connected to a 1.2 A LED driver (Thorlabs LEDD1B). Custom hardware and firmware were designed to allow the experimenter to control LED intensity and a range of light stimulus parameters. Female flies expressing *UAS-CaMPARI* were reared as described above. Custom choice caps were filled with fresh SY10% on one side and either 2% or 20% sucrose in 2% bacto agar on the other side. Groups of five flies were briefly anesthetized on ice and aspirated into the appropriate food environments. A custom clear cap was placed on top of the mocap to enclose the flies. The 405 nM LED was placed on top (~1/4 inch above the food surface) and the flies were allowed to habituate to the environment for 1 hr. After habituation, flies received UV light stimulation for 1 hr (cycling 10 s on and 10 s off, 1.0 A). Immediately following light stimulation, flies were protected from light and brains were dissected as previously described (*Wu and Luo, 2006*). Brains were mounted between a glass microscope slide and a #1.5 cover glass separated by a custom bridge in VECTASHIELD Antifade Mounting Medium (Vector Laboratories). Samples were imaged on a Nikon A1 Confocal Microscope with a 20× air objective using 488 nm and 561 nm laser lines. Z-stacks were collected at a 1–2 μm step. All treatments were mounted under the same cover slip. Image processing was performed using ImageJ (NIH), and the experimenter was blinded to treatment. Slices containing the ROI were identified and collapsed into an average intensity projection. ROIs were measured in the red and green channel after background subtraction, and a ratio of red:green fluorescence is reported.

## Statistics

For comparisons involving only one level, we used Student's t-test to detect significant differences between two treatments or one-way ANOVA followed by Tukey's post hoc test after verifying normality and equality of variances. t-Tests were two-tailed during initial characterization experiments, or one-tailed in future experiments where the predicted direction of change was known. For comparisons involving more than one level, we used two-way ANOVA to detect significant interactions between the levels and followed up with Tukey's post hoc when significance was detected (p<0.05). In cases where experimental replicates were pooled, a two-way ANOVA with blocking for experiment was performed to ensure non-significant experimental effects. For all dot and bar plots, error bars represent the SEM. All statistical tests and graphing were performed using R. Specific details of statistical analyses are presented in the figure legends.

## Acknowledgements

We wish to thank past and present members of the Pletcher laboratory for their support and comments about the experimental design of these studies, and express gratitude to David Paris for his help engineering and creating some tools used in this study. We acknowledge Binyamin Jacobovitz in the Michigan Medicine Microscopy Core for training and advice on confocal imaging. *Figure 1A* was created with BioRender.com. This research was supported by The National Science Foundation Graduate Research Fellowship Program (No. DGE 1256260) and the Howard Hughes Medical Institute through the James H Gilliam Fellowships for Advanced Study program to KJW (#GT11426) and the

US National Institute of Health, National Institute on Aging (R01 AG051649, R01 AG030593, and R01 AG063371) and the Glenn Medical Foundation to SDP.

## Additional information

### Competing interests

Scott D Pletcher: is a share holder in the company, Flidea, which has developed technology related to the FLIC feeding system. The other authors declare that no competing interests exist.

### Funding

| Funder | Grant reference number | Author |
|---|---|---|
| National Science Foundation | Graduate Research Fellowship Program DGE1256260 | Kristina J Weaver |
| Howard Hughes Medical Institute | James H Gilliam Fellowships for Advanced Study GT11426 | Kristina J Weaver |
| National Institute of Health, National Institute on Aging | R01 AG05169 | Scott D Pletcher |
| National Institute of Health, National Institute on Aging | R01 AG030593 | Scott D Pletcher |
| National Institute of Health, National Institute on Aging | R01 AG063371 | Scott D Pletcher |
| Glenn Medical Foundation | | Scott D Pletcher |

The funders had no role in study design, data collection and interpretation, or the decision to submit the work for publication.

### Author contributions

Kristina J Weaver, Conceptualization, Data curation, Formal analysis, Supervision, Funding acquisition, Investigation, Methodology, Writing – original draft, Project administration, Writing – review and editing; Sonakshi Raju, Formal analysis, Investigation, Writing – original draft; Rachel A Rucker, Formal analysis, Investigation, Methodology; Tuhin Chakraborty, Robert A Holt, Formal analysis, Investigation; Scott D Pletcher, Resources, Software, Supervision, Funding acquisition, Methodology, Writing – original draft, Project administration, Writing – review and editing

### Author ORCIDs

Kristina J Weaver ⓘ http://orcid.org/0000-0003-2496-3568
Rachel A Rucker ⓘ http://orcid.org/0000-0002-1434-9401
Scott D Pletcher ⓘ http://orcid.org/0000-0002-4812-3785

Reviewer #1 (Public Review): https://doi.org/10.7554/eLife.84537.3.sa1
Reviewer #2 (Public Review): https://doi.org/10.7554/eLife.84537.3.sa2
Author Response: https://doi.org/10.7554/eLife.84537.3.sa3

## Additional files

### Supplementary files
• MDAR checklist

## Data availability

All data generated or analyses during this study are included in the manuscript and supporting files. FLIC data were analyzed using custom R code, which is available at https://github.com/PletcherLab/FLIC_R_Code (copy archived at swh:1:rev:9713cfeb88c26e7aa4c0d87b26fcf3361500c670).

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
