## [Editor Report · eLife assessment]

This **important** paper advances our ability to understand feeding behavior in fruit flies and begins to address the challenging question of motivation. With innovative methods based on the detailed monitoring of interactions between foods of different qualities at different hunger states, they present **compelling** evidence for non-homeostatic feeding not driven by metabolic need.

---

## [Referee Report · Reviewer #1 (Public Review)]

This study aims to identify the existence of hedonic feeding and to distinguish it from homeostatic feeding, in *Drosophila*. The authors use direct observation of feeding events, a novel automated feeding event detector, inventive behavioral assays, and genetics to separate out the ways that the *Drosophila* interacts with food. Using two choice assays, the authors find an increased duration of interactions with high-concentration sugars under conditions of expected satiety, which is considered to be hedonic feeding.

The technical advances in the measurement of animal interactions with food will help advance the understanding of feeding behavior and motivational states. The correlation of specific types of food interactions across satiation state, sex, and circadian time will help drive forward the understanding of the scope of an animal's goals with feeding, and likely their relation between species and eating disorders. The assessment of mushroom body circuitry in a type of food interaction is helpful for understanding the coding of feeding control in the brain.

The bulk of the feeding data presented in the manuscript are from the interactions of individual flies with a source of liquid food, where interaction is defined as 'physical contact of a specific duration.' Although the assay they use allows for measurements to be made at high temporal resolution, the authors include some data showing that solid food consumption follows the same trend.

---

## [Referee Report · Reviewer #2 (Public Review)]

Weaver et al. used video analysis of flies that were feeding in their previously developed FLIC assay to begin to dissect the mechanisms of feeding. FLIC or Fly Liquid Interaction Counter records electrical signals that are generated when a fly touches a liquid food substrate with its legs or proboscis or both. Using video data of the liquid food interactions in the FLIC assay allowed the authors to precisely identify what a fly is doing in the feeding chamber and what the relationship is between the flies' behavior and the electrical signal recorded in the assay. This analysis produced the first detailed behavioral profile of feeding flies and allowed the authors to categorize different types of feeding in the FLIC assay, from tasting food (using their legs) to fast and long feeding bouts (using their proboscis).

After establishing what FLIC signals correspond to the different types of feeding, they used these signals to examine the food choices of starved and sated flies when presented with a sugar-rich (2% sucrose) or protein-rich (2% yeast + 1% sucrose) liquid food source. To represent hedonic feeding, they also presented flies with a choice between super sweet (20% sucrose) food or protein-rich (2% yeast + 1% sucrose) liquid food. Although fully fed flies show no difference in the number of times they visit either food choice, the flies spend more time feeding during their visits on 20% sucrose food than they do on regular sugar and on the yeast food source, suggesting that 20% sucrose is a more pleasurable food source. To make sure this was not due to the higher caloric content of 20% sucrose, they also offered flies food with the same sweetness as 20% sucrose (2% sucrose + 18% arabinose) but without caloric content and food with the same caloric content but the sweetness of 2% sucrose (2% sucrose + 18% sorbitol). This experiment showed that sweetness was the driver for the longer feeding bouts, confirming that sweeter food is apparently perceived as more pleasurable. They also looked at the effect of starving flies on the hedonic drive and found that starvation increases the time spent feeding on pleasurable food, consistent with findings in mammals that homeostatic feeding affects the hedonic drive.

To begin dissecting circuits underlying hedonic drive, the authors used CaMPARI expression in all neurons. CaMPARI is a green fluorescent reporter that turns red in the presence of Ca2+ (a measure of neuronal activity) and UV exposure. Fully fed flies in the super sweet food choice condition showed more red fluorescence in the mushroom bodies. Inhibiting a subset of these neurons acutely shows that horizontal lobes are required for the increased duration of feeding bouts on super sweet food. These lobes are innervated by a cluster of DA neurons and inhibiting them also blocks the increased super sweet feeding times.

The data in the paper largely support the conclusions. The application of this tool to distinguish between homeostatic and hedonic feeding is innovative and very compelling. As proof of the principle of the strength of their paradigm, the authors identify a distinct brain circuit involved in hedonic feeding. The methods established in the paper make a deeper understanding of feeding mechanisms possible at both a genetic and brain circuit level.

---

## [Author Response]

The following is the authors' response to the original reviews.

In brief, we incorporated all wording and clarity suggestions into the manuscript. We also updated figure legends to include additional details, including replicate numbers. New data have been added in response to requests from the reviewers. Volumetric intake data are included as a supplemental figure (Figure 1–Figure Supplement 1A) and we will include movies of the confocal stacks from our CaMPARI imaging. We worked hard to address all the reviewers’ concerns and provide a detailed response below to the reviewers’ public comments as well as their author-specific comments.

**Reviewer #1 (Public Review):**
1. All feeding data presented in the manuscript are from the interactions of individual flies with a source of liquid food, where interaction is defined as 'physical contact of a specific duration.' It would be helpful to approach the measurement of feeding from multiple angles to form the notion of hedonic feeding since the debate around hedonic feeding in *Drosophila* has been ongoing for some time and remains controversial. One possibility would be to measure food intake volumetrically in addition to food interaction patterns and durations (e.g. via the modified CAFE assay used by Ja).

We acknowledge that our FLIC assays address only one dimension of feeding behavior, physical interaction with liquid food. However, there is clear evidence that interactions are strongly predictive of consumption, and it would be technically difficult to measure feeding durations at the resolution of milliseconds using a Café assay. Nevertheless, we appreciate the spirit of this comment and agree that expanding our inference to other measures of feeding, as well as feeding environments, is an important next step. To this end, we now include measures of feeding on more traditional solid food, using the ConEx assay, and find that flies in the hedonic environment consume twice as much sucrose volume compared to flies in the control environment. These have been added as supplemental data (Figure 1 – Figure Supplement 1A), and the text has been updated to reflect our findings.

2. Some of the statistical analyses were presented in a way that may make understanding the data unnecessarily difficult for readers. Examples include:a) In Table I the authors present food interaction classifications based on direct observation. These are helpful. However, the classification system is updated or incompletely used as the manuscript progresses, most importantly changing from four categories with seven total subcategories to three categories and no subcategories. In subsequent data analyses, only one or two of these categories are assessed. It would be helpful, especially when moving from direct observation to automated categorization, to quantify the exact correspondences between all of the prior and new classifications, as well as elaborate on the types of data that are being excluded.

We appreciate the feedback on our usage of the behavioral classification system and have made several adjustments to improve it. We renamed some of the behaviors to make them more intuitive (see Reviewer #2, comment #1), and updated the main text and Table 1 to reflect these changes. We updated the text and figures to be more transparent about when we group subcategories into main categories for quantification and when we quantify all subcategories separately. Because these videos required manual scoring by an experimenter, after our initial characterizations we opted to score only main categories (which contain subcategories). We agree that it would be useful to quantify correspondence between subcategories and the automated FLIC signal. However, we believe this task is better suited for more advanced and automated video tracking software, and, incidentally, more sophisticated analysis of FLIC data, which has a very high-dimensional character that has yet to be properly exploited. At the moment, therefore, we are not confident in the ability to understand the data at the desired resolution.

b) The authors switch between a variety of biological and physiological conditions with varying assays, which makes following the train of reasoning nearly impossible to follow. For example, the authors introduce us to circadian aspects of feeding behavior to introduce the concept of 'meal' and 'non-meal' periods of the day. It is then not clear in which of the subsequent experiments this paradigm is used to measure food interactions. Is it the majority of the subsequent figure panels? However, the authors also use starved flies for some assays, which would be incompatible with circadian-locked meals. The somewhat random and incompletely reported use of males and females, which the authors show behave differently, also makes the results more difficult to parse. Finally, the authors are comparing within-fly for the 'control environment' and between flies for their 'hedonic environment' (Figure 3A and subsequent panels), which I believe is not a good thing to do.

We apologize for our difficulties conveying our inference, which was also noted by Reviewer #2. We have worked hard to improve this component in the revision. With respect to the confusion about circadian feeding, we introduced circadian meal-times to complement starvation as a second (perhaps more natural) way to measure behaviors associated with hunger. Importantly, we do not use circadian meal-times beyond Figure 1; all subsequent FLIC experiments were conducted during non-meal times of day for 6 hours, which avoids confounding our data with circadian-locked meals even when we use starved flies. We have clarified this point in the revision.

The reviewer also points out that we make both within-fly and between-fly comparisons, which is a point that we note. Perhaps some concern arises, again, from the challenges that we faced in properly delineating our inferences about different types of feeding measures (and motivations). Inference about homeostatic feeding was made using within-fly measures, comparing events on sucrose vs. those on yeast. Inference about hedonic feeding was made using between fly measures (average durations of different flies on 2% vs. 20% sucrose). Treatment comparisons to control always used measures of the same type, such that inference was not made using between-fly measures for treatment and within-fly for control (i.e., all of our figure panels were either within-fly or between fly). We have worked to clarify this in the revision.

Importantly, our approach to all experiments avoided confounding by used randomized design at multiple levels (e.g., randomizing control and hedonic environments to FLIC DFMs, alternating food choice sidedness in the DFMs), by ensuring that flies in both environments are sibling flies that came from the same vial environment before being tested, and by performing each experiment multiple times.

c) Statistical analyses are not always used consistently. For example, in Figures 3B and C, post hoc test results are shown for sucrose vs. yeast interactions, but no such statistics are given for 3E and 3F, preventing readers from assessing if the assay design is measuring what the authors tell us it is measuring.

We report p-values for two-way ANOVA interaction terms for all appropriate experiments. If (and only if) the interaction term is significant, we conduct post-hoc tests for more detailed statistical analysis and report the p-values. The reviewer points out that we do not perform post-hoc tests in figures 3E and 3F. These figures had a non-significant interaction term, and thus, we did not feel a post-hoc test was warranted.

**Reviewer #2 (Public Review):**
1. The dissection of feeding into distinct behavioral elements and its correlation with electrical FLIC signals that allow interpreting feeding types is a fundamental new method to dissect feeding in flies. However, the categories of micro-behaviors in Table 1 are not intuitive.

We agree and have updated the Table, figures, and main text. Please see also our response to Reviewer #1, comment #1.

2. The details for the behavioral data analysis are not clear and should be made more obvious. For example, how many males and females were used in each experiment? Were any of the females mated or were they all virgins? If all virgins, why not use mated females? Mating status may have an effect on the feeding drive. If mated and virgin females were used, are there any differences between them? Similarly, for diurnal feeding experiments, it is not immediately clear from the graphs how many animals were used and how the frequencies were obtained (Fig. 1F, presumably averages for each category per fly but that is inconsistent with the legend in the supplement for this figure). Why does the transition heat map not include all micro-behaviors (Fig. 1E, no LQ data which are significant in diurnal feeding)?

We have clarified the number of flies and events for each behavioral experiment in Figure 1, and we updated the figure legend appropriately. We note that these behavioral datasets are non-overlapping, and each time we mention the number of events scored in the text, that number includes only “new” videos. Female and male flies for all experiments were mated, and we have clarified this in the main text and methods.

For the diurnal experiment in Figure 1F, we scored over 700 events from new (non-overlapping) video compilations and updated the number of flies and event number in the figure legend. The diurnal data we present in the supplement for this figure is a separate experiment conducted on 38 flies, intended only to demonstrate the circadian nature of fly feeding.

For the transition heat map, analysis of this sort seems to require a large amount of data to have sufficient power to return a transition matrix. LQ events are relatively low in frequency, so we opted to combine them with L events for this analysis. We have updated the figure and figure legend to reflect this.

3. The CaMPARI images do not look great, particularly in the pan-neuronal condition (Fig. 5A). It would be useful to include the movie of the stack. Did any other brain regions show activity differences, such as SEZ or PI? These regions are known to be involved in feeding so it seems surprising they show no effect.

We find that CaMPARI imaging is subject to high levels of noise and background, especially when using a broad driver as the reviewer has pointed out. This is why we opted to follow-up our pan-neuronal CaMPARI experiment using a more specific mushroom body driver and to test our correlational findings of increased MB activity in hedonic environments with genetic approaches in the remainder of Figure 5. We have included movies of the confocal stacks for both CaMPARI experiments, as requested.

**Reviewer #1 (Recommendations For The Authors):**

Main concern:No measurements of intake, either in volume or in caloric value. Hence, 'hedonic' feeding is only indirectly supported.I would like to suggest to the authors that they measure intake volumetrically in addition to food interaction patterns and durations. For example, William Ja developed a modified CAFE assay that measures consumption volume in real-time in freely behaving flies (http://dx.doi.org/10.1038/nprot.2017.096). Liming Wang has another capable assay. Additional values of expanding measurement methods for feeding are that it helps tie the research more directly to that of others, and it helps remove the concern that any one assay may introduce unknown biases.For the CaMPARI, it would be helpful to provide a demonstration of its effectiveness by recapitulating a deep brain neural pathway known to be engaged by a stimulus by GCaMP or electrophysiology.Additional concern:The authors assume satiety states during different circadian periods (line 253, for example). It seems critical to directly measure the satiety state.Technical concerns:Figure 5 A, B: there is reported near zero UV transmission through the head: https://doi.org/10.1364%2FBOE.6.000514, hence the CaMPARI measurements are suspect. It appears that there may be an effect in the optic lobes that may receive greater UV illumination by being more peripheral. A positive control to demonstrate deep brain access by UV is needed.Y-axes vary for the same measurement types within figures, for example, Figure 5 C-G. Also Figures 3F, G, I, K, M and Figures 3D, E, H, J, L. This hinders direct comparisons.Figure 2: why are there no statistics to distinguish interaction (I) events from F and L? Why are the example graphs presented using different scale x-axes? For A-C, why no averaged response graphs for the classifications? Were there other events that did not fit these classifications?In lines 224-226, the claim of statistical significance at p=0.061 makes the reader suspicious of the statistical interpretations throughout the manuscript.Figure 3B starved looks the same as Figure 3C sated for females, using the same assay and conditions. This implies a huge amount of variance in behavior between experiments.

We appreciate the recommendations from Reviewer #1 and have done our best to address many of their concerns. Regarding their main concerns, we have added volumetric feeding data to the manuscript, included movies of the confocal stacks for the CaMPARI experiments, and clarified the circadian timing of our behavioral experiments. These details are outlined in our public response to both reviewers. The reviewer also expressed a few technical concerns, mostly regarding statistical analyses. We agree that there seems to be a large amount of biological variability between experiments, which we do indeed find to be the case with behavioral experiments of this sort. For this reason, we avoid making direct comparisons on absolute values between experiments, as the reviewer suggests, and thus allow our Y-axes to vary for each figure to better facilitate within-experiment comparisons. The reviewer also points out that, in one instance, we refer to a p-value of 0.061 as statistically significant in the text. While we have changed our language to reflect the perceived convention, we note that there is little inferential difference between these values, and we report exact p-values to allow the reader to make an informed decision.

**Reviewer #2 (Recommendations For The Authors):** The writing and data presentation in this paper is somewhat dense and confusing at times. Comments and questions below are intended to help improve data presentation and resolve questions that will help the reader navigate and understand the data to better appreciate the significance of the findings.Comments and questions:Line 160 cites Chen et al, 2002 as an example of behavioral characterization that is useful for read-outs of neural states, but no neural states were defined in that work. A better example where a circuit was linked to a specific behavioral category is PMID30415997 (Duistermars et al., 2018).Line 171: were the females mated or virgin or was it variable?The classification system in Table 1 is a bit confusing. For example, the distinction is made between Fast and Long feeding events as well as interactions with food and other events. FH meet the requirements of F and H, presumably meaning that flies are fast feeding and touching the food with their front legs. Why are front legs and hind legs touching food abbreviated H and FF respectively instead of something more obvious like IF and IH (referring to Interaction with Front legs or Interaction with Hind legs)?Also was there never any tasting with the middle legs? In Fig1B, all the I events are grouped. Are most of these H or FF events? The frequency in Fig. 1B is shown as normalized as a frequency of all events. The statistical analyses are all parametric. Are these data normally distributed?Lines 224-229: the relative frequency of L-type feeding is increased in starved flies and the relative frequency of F feeding is decreased. Is the relative L- or F-type feeding frequency considered on total behavior or just the sum of long and fast feeding or the sum of all types of feeding?The events that are analyzed vary throughout the paper. Line 173 mentions 300 events, line 222, 500 events, and line 257, 700 feeding events. Are these all independent experiments, or are these overlapping data sets analyzed for different parameters?For diurnal feeding behavior, the authors analyzed 700 events and found significantly more LQ events during meal time (i.e. at the beginning and end of the day). Based on the figure legend in the supplement to Figure 1, it appears that these data were collected on 38 female flies. But in Fig 1F, there are ~8 points per feeding type (F, L, and LQ) during meal and non-meal conditions. Shouldn't all 38 flies have an average frequency for each type of feeding during meal and non-meal times? Were these females mated or not? Is this effect also true for males? To help the reader understand the data better, it would be helpful to note the number of flies used in each experiment or in each analysis in the different figures and wherever the data are mentioned in the manuscript. It also seems likely that the mating state may have an effect on feeding so knowing the result in mated versus unmated would be a useful analysis.It is interesting that there is a difference in feeding in starved flies versus diurnal feeding in the presumably hungry versus sated phase (meal versus non-meal phases). As mentioned by the authors earlier in the manuscript, starved flies have a relative increase in L-type feeding. However, they perform less LQ feeding than sated flies, and yet LQ feeding is the only significantly different type of feeding in the hungry state of diurnal feeding. In the morning, the transition to feeding is very abrupt compared to the gradual increase in the evening. Is there any difference between the type of feeding or the transition matrix in the evening versus morning meal times? Also, why is LQ feeding not included as a category in the transition matrix in Fig 1E?In Fig 2, the authors examine FLIC signals with video data to identify feeding types from FLIC signals. Why are there signal durations for F-type feeding that are longer than 3 seconds when it is defined as 1-3 sec of the proboscis contact with food and conversely signals of L-type feeding shorter than 4 seconds when it is defined as >4 seconds of continuous proboscis contact? Does this mean that signal can be longer or shorter than the actual time the proboscis is in the food?With these parameters, the authors develop an assay to identify homeostatic and hedonic feeding by applying the signal analysis to food choices representing homeostatic (2% sucrose versus yeast) and hedonic (2% sucrose versus 20%) conditions. In Fig 3C, they show that fully-fed females show a stronger preference for yeast food than sugar food compared to males (line 335). Is this in fully fed animals? The yeast preference in females looks almost the same as in the starved females in Fig 3B.The CaMPARI images shown in Fig 5A (and to a lesser extent Fig 5B) are not particularly convincing although the quantification looks clear. Providing the movies of the stacks may help the reader better appreciate the difference in MB red signal in the hedonic state. It would also help to show the number of flies that were tested in these experiments as well as the sex and mating status. Provide the n in the figure legend and in the relevant sections in the text.Were the mushroom bodies the only brain region with significant, measurable activity changes? One might expect changes in other feeding areas, such as the subesophageal zone (SEZ) and the peptidergic regions of the brain (PI), which are both known to affect feeding in flies. This may also be a useful method to examine differences in mated versus unmated flies.In Fig 5C the caption reads MB lambda lobe inhibition. Shouldn't this be gamma lobe inhibition as suggested in the figure legend?The paper largely distinguishes homeostatic from hedonic feeding only. It may be useful to discuss other non-homeostatic mechanisms as well or at least make the distinction in the introduction and or discussion.

We thank reviewer #2 for their thoughtful suggestions to improve the clarity of the manuscript. They suggest several improvements, which we implemented, including that we improve the classification system in Table 1 to make it more intuitive, state how we normalized observed behavioral frequencies, clarify that the number of events we cite for each experiment are non-overlapping, and explain the use of circadian meal vs. non-meal times. We also noticed, as did this reviewer, that the usage of L vs. LQ events differs between starved flies and flies observed during meal-time. We agree that it may be interesting to sort out the nuances of why and how these differences occur, as it suggests that starvation may in some ways be different from physiological hunger. However, our method of manually observing flies would make this difficult at present. We hope to utilize more advanced video tracking software in the future to investigate this question. The reviewer also posed several questions about the hunger/satiety state of flies that we used for each experiment, which we clarified throughout the main text, figure legends, and methods.

This reviewer points out two technical concerns, which we have addressed. The concerns about our CaMPARI imaging are noted, and we have discussed them in response to reviewer #1 and in our public response. We now include movies of the confocal stacks, as requested. There was also a question about FLIC durations of F and L events in Figure 2, with some visually identified F events producing FLIC signals longer than 4 seconds and some L events producing FLIC signals shorter than 4 seconds. Although we show that population averages from the FLIC can reliably recapitulate our visual metrics, there is occasional noise at the individual level. For example, although a fly may have contact of its proboscis with the food for less than 4 seconds, the FLIC signal may persist slightly beyond that interaction due to sustained contact with a non-proboscis body part or due to liquid food contacting the signal pad. We also occasionally observed L events that we visually identified to last longer than 4 seconds, but nevertheless did not produce a FLIC signal of equal length. This can occur when a fly feeds on the liquid food but transiently loses contact with the signal pad. Although there is some noted technical noise, we show that population-level data is sufficient to reflect our visual observations.